# Fibrosis-4 Score Is Associated with Mortality in Hemodialysis Patients with Chronic Viral Hepatitis: A Retrospective Study

**DOI:** 10.3390/diagnostics14182048

**Published:** 2024-09-15

**Authors:** Hao-Hsuan Liu, Chieh-Li Yen, Wen-Juei Jeng, Cheng-Chieh Hung, Ching-Chung Hsiao, Ya-Chung Tian, Kuan-Hsing Chen

**Affiliations:** 1Department of Nephrology, New Taipei Municipal TuCheng Hospital, New Taipei City 236043, Taiwan; hhliu@cgmh.org.tw (H.-H.L.); b9002024@cgmh.org.tw (C.-C.H.); 2Kidney Research Center, Chang Gung Memorial Hospital, School of Medicine, Chang Gung University, Taoyuan 333423, Taiwan; b121919702@cgmh.org.tw (C.-L.Y.); cchung@cgmh.org.tw (C.-C.H.); dryctian@cgmh.org.tw (Y.-C.T.); 3Department of Gastroenterology and Hepatology, Chang Gung Memorial Hospital, Linkou Branch, Taoyuan 333423, Taiwan; m7197@cgmh.org.tw; 4College of Medicine, Chang Gung University, Taoyuan 333323, Taiwan

**Keywords:** FIB-4 score, end-stage kidney disease, hemodialysis, chronic viral hepatitis, mortality, major adverse cardiovascular events

## Abstract

BACKGROUND: Chronic hepatitis B and C infections are major causes of morbidity and mortality in end-stage kidney disease (ESKD) patients on hemodialysis (HD). The Fibrosis-4 (FIB-4) score is a non-invasive method to evaluate chronic liver disease. However, it is unclear whether there is a connection between the FIB-4 score and major adverse cardiovascular events (MACEs) and mortality in patients on HD. This study investigates the relationship between FIB-4 scores, MACEs, and mortality in HD patients. METHODS: A 5-year retrospective study included 198 HD patients with chronic hepatitis B and C from Chang Gung Memorial Hospital. FIB-4 scores were categorized into high (>2.071), middle (1.030~2.071), and low (<1.030) tertiles for cross-sectional analyses. MACEs and mortality were tracked longitudinally. RESULTS: Patients with high FIB-4 scores had lower hemoglobin and albumin levels. Cox multivariate analysis showed that high FIB-4 scores (aHR: 1.589) and diabetes mellitus (aHR: 5.688) were significant factors for all-cause mortality. The optimal FIB-4 score for 5-year mortality was 2.942. FIB-4 scores were not significant for predicting 5-year MACEs. CONCLUSIONS: High FIB-4 scores are associated with increased 5-year all-cause mortality risk in HD patients with chronic hepatitis virus infection.

## 1. Introduction

Chronic viral hepatitis is a significant threat to individuals with end-stage kidney disease (ESKD), especially those with liver cirrhosis. It substantially reduces their quality of life and increases morbidity and mortality [1]. Among ESKD patients undergoing hemodialysis (HD), chronic hepatitis C (CHC) infection prevalence ranges from 9% to 80% [1,2,3], while chronic hepatitis B (CHB) infection varies between 1% and 15.3% worldwide [4,5].

Liver cirrhosis is a serious consequence of chronic viral hepatitis. Biopsy is the gold standard for diagnosis, but it is invasive and risky, potentially leading to complications like bleeding, bile leakage, infection, pneumothorax, intestinal perforation, and even mortality [6]. Notably, patients with ESKD face a higher risk of bleeding due to impaired platelet function [7,8]. Furthermore, liver biopsy is linked to diagnostic errors in disease staging, with a potential error rate of up to 20% [9].

Therefore, non-invasive alternatives to biopsy have been proposed. These include imaging-based approaches like Magnetic Resonance Elastography (MRE) [10] and Transient Elastography (TE) using instruments such as FibroScan^®^ [11], as well as blood-based tests like the AST/ALT ratio (AAR), AST-to-Platelet Ratio Index (APRI), Fibrosis-4 (FIB-4) score, and Fibrosis Index [12,13,14,15]. Nevertheless, these methods face limitations in terms of accessibility. MRE is characterized by its high cost, limited availability, and less-than-optimal sensitivity, while TE is influenced by operator skills. In patients with ESKD, the validity of TE is contingent on the timing, as it becomes reliable only after HD due to potential confounding factors like increased liver density resulting from parenchymal congestion due to venous hyperemia in liver sinusoids, as well as edema and/or ascites [16].

Apart from using instrumental techniques for histological and functional liver assessment, various liver fibrosis indices (LFIs) have been formulated. These LFIs are derived from clinical data or alterations in markers that either directly or indirectly reflect the extent of fibrotic liver tissue damage.

In the last decade, numerous new LFIs have emerged, with some finding clinical utility, especially among individuals with viral liver diseases and nonalcoholic fatty liver disease. For instance, the FIB-4 score was created as a non-invasive tool to assess liver disease progression in individuals with coinfection of human immunodeficiency virus and hepatitis C virus [14]. The FIB-4 score was calculated based on the following formula: FIB-4 = [age (years) × AST (IU/L)]/[platelet count (10^9^/L) × ALT (IU/L)^1/2^]. Prior research has shown the diagnostic precision and predictive value of the FIB-4 score in evaluating the degree of liver fibrosis and steatosis among individuals receiving HD [17,18,19,20,21]. Nevertheless, the relationship between the FIB-4 score and the occurrence of major adverse cardiovascular events (MACEs) and mortality in HD patients remains unexplored. This study aims to examine the potential of the FIB-4 score to predict MACEs and mortality in chronic viral hepatitis patients undergoing HD.

## 2. Materials and Methods

### 2.1. Patient Recruitment

Between January 2011 and January 2016, a total of 310 patients with chronic viral hepatitis infection (CHB: 130 patients, CHC: 159 patients, and CHB-CHC coinfection: 21 patients) who received HD at Chang Gung Memorial Hospital were retrospectively recruited. The inclusion criteria were as follows: 1. ESKD treated with HD for at least 6 months; 2. serology positivity for hepatitis B surface antigen (HBsAg) or anti-hepatitis C virus (anti-HCV) antibody for at least 6 months; 3. follow-up for at least 60 months. Those with history of MACEs, HCC, or hepatic decompensation (defined as patients with jaundice, ascites, hepatic encephalopathy, hepatorenal syndrome, or variceal hemorrhage) [22] prior to the enrollment, and those who received antiviral treatment for CHB or CHC prior to or during the enrollment, were excluded.

Personal history, including history of tobacco use, alcohol use, or drug use, was documented by chart review. Primary renal disease, comorbid conditions and the primary outcome, and the presence of MACEs, including coronary arterial disease (CAD), congestive heart failure (CHF), cardiovascular death (CV death), and cerebrovascular disease (CVD) [23], were identified after an in-depth review of medical records, including history and physicals, progress notes, discharge summaries, consults, and biopsy results. The cause of death, including infection, cardiovascular, intracerebral hemorrhage, or massive gastrointestinal bleeding events during follow-up, were obtained from the discharge diagnosis and death certificates were obtained from hospital records.

### 2.2. Laboratory Methods

All laboratory values, including blood cell counts, biochemical data, the urea reduction ratio (URR), the normalized protein catabolic rate (nPCR), the time-averaged concentration of blood urea nitrogen (TACurea), and renal KT/V Daugirdas urea, were measured by automated and standardized methods. All of the blood samples of the patients (after fasting for at least 12 h) were collected, centrifuged, and then immediately examined. Serum white blood cell (WBC) counts, hemoglobin (Hb), platelets, aspartate aminotransferase (AST), alanine aminotransferase (ALT), total bilirubin, γ-glutamyltransferase (γ-GT), alkaline phosphatase, albumin, glucose, glycated hemoglobin A1c (HbA1c), blood urea nitrogen (BUN), creatinine (Cr), uric acid, sodium, potassium, phosphate, aluminum, iron, total iron-binding capacity (TIBC), ferritin, intact parathyroid hormone (i-PTH), high-sensitivity C-reactive protein (hs-CRP), total cholesterol, triglyceride, high-density lipoprotein (HDL), and low-density lipoprotein (LDL) were assayed by standard central automated laboratory methods. Serum calcium was corrected for serum albumin according to the following formula: corrected calcium (mg/dL) = serum calcium (mg/dL) + 0.8 × [4.0 − serum albumin (g/dL)].

### 2.3. Statistical Analysis 

Continuous variables were expressed as means ±SD and analyzed by one-way ANOVA. Categorical variables were expressed as a number or percentage for each item and analyzed by the Fisher exact test or Chi-square test according to the number. The Cox proportional hazard model was applied for the predictors of MACEs and mortality. Factors which reached *p* < 0.1 in the univariate analysis were entered into the multivariate analysis model by the “conditional forward stepwise” method. The predictive accuracy of the FIB-4 score for identifying all-cause 5-year mortality in ESKD patients with chronic viral hepatitis was assessed by the area under the receiver operating characteristics (AUCs). The Youden index was used to find the optimal cut-off with the greatest sensitivity and specificity, which were associated with 46.88% and 93.94% [24]. The diagnostic accuracy was evaluated by calculating the sensitivity, specificity, positive predictive values (PPVs), and negative predictive values (NPVs). A two-tailed *p*-value of less than 0.05 was considered significant. All statistical calculations were performed with SPSS 19.0 for Windows (IBM Corporation, Armonk, NY, USA).

## 3. Results

### 3.1. Patient Characteristics

A total of 198 HD patients with chronic viral hepatitis who met the inclusion and exclusion criteria were analyzed (Figure 1). Clinical baseline characteristics are listed in Table 1; the mean age was 56.91 ± 11.72 years old, 53.03% were male, 47.98% were CHB virus carriers, 47.47% were CHC-infected patients, and the rest of the 4.55% were CHB/CHC coinfections. The mean FIB-4 score was 1.80 ± 1.30. The mean HD duration at baseline was 10.26 ± 8.17 years. Table 1 shows the characteristics of patients in the three subgroups according to the tertile of the FIB-4 score. The high-FIB-4 group (mean FIB-4 = 3.23 ± 1.29, *n* = 66) had a significant trend of a higher age, cardiothoracic ratio, AST, and ferritin, as well as lower WBC counts, Hb, platelets, albumin, BUN, Cr, potassium, phosphate, TIBC, total cholesterol, triglyceride, and LDL compared to the other groups (middle-FIB-4 group: mean FIB-4 = 1.47 ± 0.28, *n* = 66; low-FIB-4 group: mean FIB-4 = 0.72 ± 0.21, *n* = 66). The three groups had a comparable proportion of gender, body weight, presence of diabetes mellitus (DM) or hypertension, HD duration, URR, KT/V Daugirdas, nPCR, TACurea, ALT, glucose, HbA1c, uric acid, sodium, corrected calcium, aluminum, iron, hs-CRP, and HDL.

### 3.2. Predictors for 5-Year MACEs in HD Patients with Chronic Viral Hepatitis

During the 5-year follow-up, 21 patients (10.61%) had MACEs in HD patients with chronic viral hepatitis. The incidence of MACEs in the 5-year follow-up showed no difference among the three study groups (3.54% vs. 3.54% and 3.54%, respectively). Among the MACEs, CAD accounted for 42.86%, followed by CVD (23.81%), CHF (19.05%), and CV death (14.29%), respectively.

All of the variables listed in Table 1—including age, male sex, body weight after HD, presence of DM, presence of hypertension, HD duration, cardiothoracic ratio, URR, KT/V Daugirdas, nPCR, TACurea, WBC counts, Hb, platelets, AST, ALT, total bilirubin, albumin, glucose, HbA1c, BUN before HD, creatinine, uric acid, sodium, potassium, corrected calcium, phosphate, aluminum, iron, TIBC, ferritin, i-PTH, hs-CRP, total cholesterol, triglyceride, HDL, LDL, and the FIB-4 score—were used to identify the possible predictors of 5-year MACEs through a univariate Cox regression analysis. Subsequently, all significant variables (*p* < 0.05) from the univariate analysis—including body weight, presence of DM, HD duration, URR, KT/V Daugirdas, TACurea, WBC counts, total bilirubin, glucose, corrected calcium, and HDL—were included in the multivariate Cox regression analysis to adjust for the predictors of 5-year MACEs. Using the multivariate Cox regression analysis, type II DM (aHR: 6.455; 95% CI: 2.466–16.893, *p* < 0.001), TACurea (aHR: 1.056; 95% CI: 1.010–1.104, *p* = 0.016), and total bilirubin (aHR: 19.360; 95% CI: 2.619–143.086, *p* = 0.004) were the significant risk factors for the 5-year MACEs of HD patients with chronic viral hepatitis after adjusting for related variables. However, the FIB-4 score was not a significant risk factor (HR: 1.222; 95% CI: 0.865–1.728; *p* = 0.256) for the 5-year MACEs of HD patients with chronic viral hepatitis (Table 2).

### 3.3. Predictors for 5-Year Mortality in HD Patients with Chronic Viral Hepatitis

By the end of the follow-up, 32 patients (16.16%) had died within the 5-year follow-up in HD patients with chronic viral hepatitis. Twenty (10.10%) patients in the high-FIB-4 group, five (2.53%) patients in the middle-FIB-4 group, and seven (3.54%) patients in the low-FIB-4 group passed away during this period. Fourteen patients died of infection, three patients were cardiovascular deaths, and three patients died of massive gastrointestinal bleeding in the high-FIB-4 group. Three patients were cardiovascular deaths, one patient died of infection, and one patient died of intracerebral hemorrhage in the middle-FIB-4 group. Four patients died of infection and three patients were cardiovascular deaths in the low-FIB-4 group. No difference in the causes of death among the three study groups was noted. Among the all-cause mortality, infection accounted for 59.38%, followed by CV death (28.13%), massive gastrointestinal bleeding (9.38%), and intracerebral hemorrhage (3.13%), respectively.

Similarly, all of the variables listed in Table 1 were evaluated using univariate Cox regression analysis for predicting the all-cause 5-year mortality. Subsequently, all significant variables (*p* < 0.05) from the univariate analysis—including age, presence of DM, Hb, AST, total bilirubin, albumin, glucose, BUN, creatinine, sodium, potassium, corrected calcium, phosphate, TIBC, ferritin, hs-CRP, total cholesterol, HDL, and the FIB-4 score—were included in the multivariate Cox regression analysis to adjust for the predictors of all-cause 5-year mortality. Using the multivariate Cox regression analysis, the FIB-4 score (aHR: 1.589; 95% CI: 1.262–2.001; *p* < 0.001) along with type 2 DM (aHR: 5.688; 95% CI: 2.358–13.720, *p* < 0.001), the Hb level (aHR: 0.524; 95% CI: 0.369–0.745, *p* = < 0.001), and the albumin level (aHR: 0.538; 95% CI: 0.296–0.978, *p* = 0.042) were the significant factors associated with the all-cause mortality of HD patients with chronic viral hepatitis after adjusting for related variables (Table 3).

### 3.4. Diagnostic Performances of FIB-4 Score for Predicting All-Cause 5-Year Mortality in HD Patients with Chronic Viral Hepatitis

The area under the ROC curve (AUC = 0.739; 95% CI: 0.633–0.845; *p* < 0.001; Figure 2) shows that the FIB-4 score has the predictive ability to discriminate all-cause mortality during the 5-year follow-up of healthy subjects. The cut-off point is 2.942 (sensitivity = 0.469; specificity = 0.939; PPV = 0.886; NPV = 0.639) and is an optimal diagnostic point of the FIB-4 score for 5-year mortality. The 5-year cumulative mortality rate was 60.01% in patients with a FIB-4 ≥ 2.942, which was much higher than the 9.88% in those with a FIB-4 < 2.942 (Figure 3: K-M plot with two groups, log rank test, *p* < 0.001).

## 4. Discussion

This is the first study to investigate the FIB-4 predictability for MACEs and mortality in chronic viral hepatitis-infected patients undergoing HD. Not surprisingly to disclose, there is no association between the FIB-4 value and the incidence of MACEs in this group of patients, since the FIB-4 level indicates the probability of cirrhosis, which is not a direct factor for MACEs. However, the FIB-4 score could serve as a common and inexpensive method for predicting all-cause mortality in HD patients with chronic viral hepatitis.

The FIB-4 score is a commonly used LFI to predict and correlate significant fibrosis and cirrhosis in CHB [25,26] or CHC [27,28] patients. According to the tertiles of the FIB-4 scores in our study, the high-FIB-4 group had a significant trend of lower platelets (*p* < 0.001), WBC counts (*p* < 0.001), and Hb (*p* = 0.017) levels compared to the other groups. Chronic liver disease is frequently associated with hematological abnormalities. Thrombocytopenia (defined as platelets ≤ 150,000/mm^3^) is the most common and first of the abnormal hematologic indices to occur in patients with cirrhosis, followed by leukopenia (defined as a WBC count ≤ 4000/mm^3^) and anemia (defined as Hb ≤ 13.5 g/dL for men and 11.5 g/dL for women). A combination of leukopenia and thrombocytopenia at baseline predicted increased morbidity and mortality in these patients [29]. The major mechanisms for thrombocytopenia in liver cirrhosis are (1) platelet sequestration in the spleen and (2) decreased production of thrombopoietin in the liver [30]. It was found that the sera of liver cirrhosis patients with leukopenia suppressed the in vitro colony formation of normal marrow CFU-C (granulocyte–macrophage colony-forming unit), and that the degree of suppression was well correlated with the severity of granulocytopenia. The appearance of this inhibitor may be one of the causes of leukopenia in liver cirrhosis [31]. Anemia of diverse etiology occurs in about 75% of these patients. The causes of anemia include acute or chronic gastrointestinal hemorrhage and hypersplenism secondary to portal hypertension [32,33,34]. Significant hypoalbuminemia was found in the high-FIB-4 group compared to the other groups (*p* = 0.003). Albumin is synthesized exclusively in the liver. Albumin levels fall as the synthetic function of the liver declines with worsening cirrhosis. Thus, serum albumin levels can be used to help grade the severity of cirrhosis [35]. The high-FIB-4 group also had a significant trend of a higher ferritin level (*p* = 0.001) compared to the other groups. Ferritin is the major protein implicated in the constitution of body iron stores. High ferritin is mostly encountered either in iron overload states due to excessive synthesis or due to increased release from damaged cells. A number of studies in the recent past have shown that patients with liver disease, including cirrhosis, have high serum ferritin levels because of hepatic necro-inflammation and the release of ferritin from damaged hepatocytes, or due to secondary to macrophage activation [36,37].

In this study, the presence of DM was regarded as the risk factor for both MACEs (HR: 6.455; 95% CI: 2.466–16.893; *p* < 0.001) and all-cause mortality (HR: 5.688; 95% CI: 2.358–13.720; *p* < 0.001). In another systematic review and meta-analysis, DM was associated with an increased risk of cardiovascular events and all-cause mortality in HD patients [38]. Increased Hb (HR: 0.524; 95% CI: 0.369–0.745; *p* < 0.001) and decreased serum albumin levels were both associated with reduced risks of all-cause mortality. This finding may result from the low Hb level always indicating a protein–energy-wasting status, which is a risk factor for mortality in dialysis-dependent populations [39,40,41,42]. Many studies suggested the association between higher Hb levels and improved quality of life, physical function, exercise capacity, and better survival [43,44,45]. Hypoalbuminemia can be contributed to various conditions, such as nephrotic syndrome [46], heart failure [47], liver disease [48], and malnutrition [49]. Most cases of hypoalbuminemia are caused by acute and chronic inflammation responses [50]. Moreover, another strong association has been reported between the serum albumin level and mortality [51].

In our study, the FIB-4 score was a significant risk factor for 5-year mortality in HD patients after adjusting for other confounding factors. In previous studies, a high FIB-4 score has consistently been associated with the increased risk of cardiovascular- and liver-related mortality [52,53,54]. It is hypothesized that chronic liver disease may be causally linked to mortality with the following possible mechanisms: (1) a mechanism through the development of arteriosclerosis [55,56,57,58], which concomitantly enhances the RAS system and deactivates nitric oxide synthesis [59,60]; (2) a mechanism mediated by liver-derived inflammatory mediators and oxidative stress [61,62]. In addition, the extent of liver fibrosis may indirectly reflect the accumulated amount of inflammatory burden in non-liver diseases such as heart failure [63]. We assume that the FIB-4 score at diagnosis could predict all-cause mortality in HD patients with chronic viral hepatitis in a similar manner. The association between the FIB-4 score and all-cause mortality has also been described in prior studies, such as patients with nonalcoholic fatty liver disease (NAFLD) [64], antineutrophil cytoplasmic antibody (ANCA)-associated vasculitis (AAV) [65], and even coronavirus disease 2019 [66]. Although these studies demonstrate that the FIB-4 score is a significant predictor for mortality in different populations, our study is the first to demonstrate an association between the FIB-4 score and all-cause mortality in HD patients with chronic viral hepatitis.

In our study, a FIB-4 score with a cut-off value ≥ 2.942 had an acceptable diagnostic accuracy (AUC = 0.739), it accurately identified the mortality in ESKD patients with chronic viral hepatitis, and it exhibited a high specificity (93.94%) and high PPV (88.55%) with a relatively low sensitivity (46.88%) and NPV (63.88%), which means that the precision is high. The 5-year cumulative mortality rate was 60.01% in patients with a FIB-4 ≥ 2.942, which was much higher than the 9.88% in those with a FIB-4 < 2.942.

This investigation has some limitations. First, since a liver biopsy was not performed, the correlation between the FIB-4 score and the actual degree of fibrosis was unknown. Concurrent laboratory assessment and liver biopsy is ideal but was not possible to control retrospectively. Second, this study was comprised of only one race from a uni-center study, suggesting a possibility of selection bias. Third, the viral load of chronic hepatitis was not taken into account in this study. However, non-invasive liver tests have never been comprehensively studied in this population for mortality, ensuring the novelty and importance of the results. Further investigation is needed to confirm our observations in the future.

## 5. Conclusions

In conclusion, the current study is the first to demonstrate that the FIB-4 score predicts 5-year all-cause mortality in HD patients with chronic viral hepatitis. The 5-year cumulative mortality rate was significantly higher in ESKD patients with chronic viral hepatitis who had a FIB-4 score ≥ 2.942, indicating that it is a risk factor for mortality. The FIB-4 score could serve as a common and inexpensive method for predicting all-cause mortality in HD patients with chronic viral hepatitis.

## Figures and Tables

**Figure 1 diagnostics-14-02048-f001:**
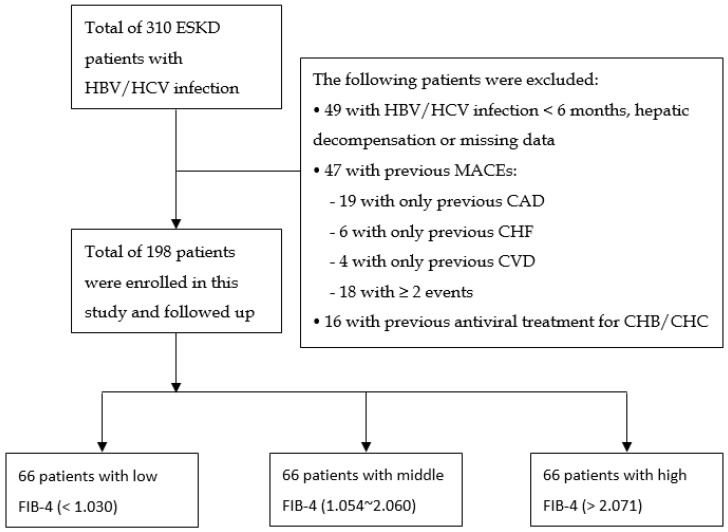
Flowchart of included patients in this study.

**Figure 2 diagnostics-14-02048-f002:**
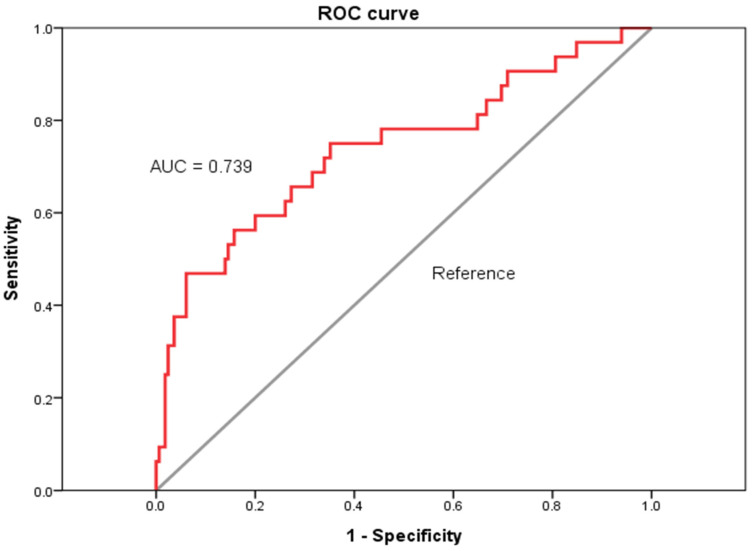
ROC curve and AUC of the FIB-4 score for identifying 5-year mortality. ROC curve: receiver operating characteristic curve; AUC: area under the curve; FIB-4 score: Fibrosis-4 score.

**Figure 3 diagnostics-14-02048-f003:**
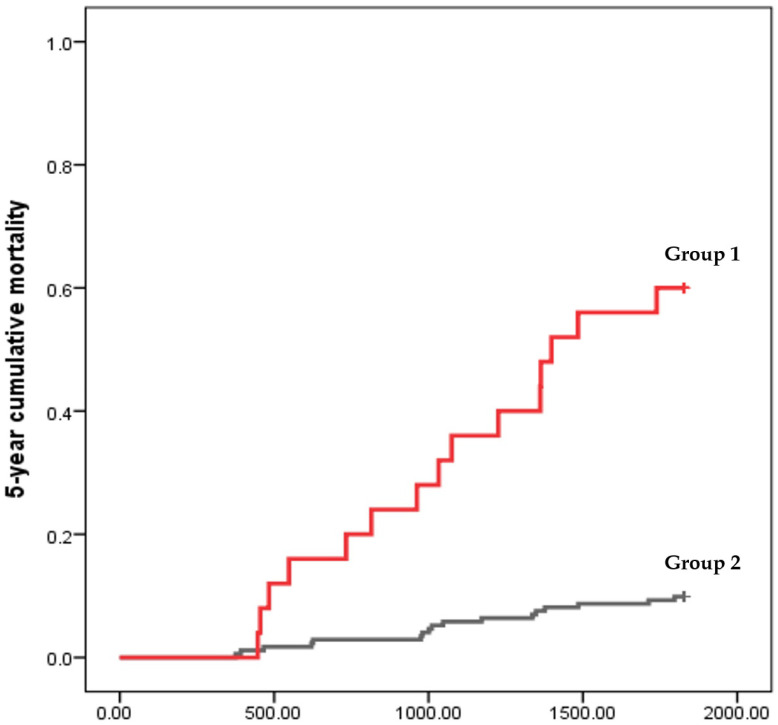
Kaplan–Meier plot with the diagnostic point of the FIB-4 score (2.942) for identifying 5-year mortality. Group 1: FIB-4 score ≥ 2.942; Group 2: FIB-4 score < 2.942.

**Table 1 diagnostics-14-02048-t001:** Clinical features of the patients stratified across the tertiles of the FIB-4 score (*n* = 198).

Characteristics	Overall	Low-FIB-4 Group (<1.030) (*n* = 66)	Middle-FIB-4 Group (1.030~2.071) (*n* = 66)	High-FIB-4 Group (>2.071) (*n* = 66)	*p*
Demographics					
Age, year	56.91 ± 11.72	49.94 ± 10.80	57.50 ± 10.39	63.38 ± 9.97	<0.001
Male sex	105 (53.03)	42 (63.64)	32 (48.48)	31 (46.97)	0.118
Body weight after hemodialysis, kg	57.68 ± 12.19	60.24± 12.36	57.05 ± 11.86	55.71 ± 12.09	0.093
Comorbidity					
Diabetes mellitus, yes	58 (29.29)	21 (31.82)	22 (33.33)	15 (22.73)	0.381
Hypertension, yes	121 (61.11)	41 (62.12)	39 (59.09)	41 (62.12)	0.887
Dialysis-related data					
Hemodialysis duration, year	10.26 ± 8.17	10.15 ± 7.63	10.81 ± 8.13	9.82 ± 8.80	0.778
Cardiothoracic ratio, %	0.50 ± 0.07	0.49 ± 0.06	0.48 ± 0.05	0.53 ± 0.08	<0.001
Urea reduction ratio (URR), %	0.77 ± 0.07	0.77 ± 0.08	0.78 ± 0.07	0.78 ± 0.07	0.809
KT/V Daugirdas	1.83 ± 0.40	1.81 ± 0.38	1.85 ± 0.44	1.83 ± 0.37	0.829
Normalized protein catabolic rate (nPCR), g/(kg·d)	1.21 ± 0.48	1.17 ± 0.39	1.20 ± 0.46	1.27 ± 0.57	0.442
Time-averaged concentration of blood urea nitrogen (TACurea), mmol/L	38.22 ± 10.35	39.63 ± 9.71	38.36 ± 10.21	36.62 ± 11.05	0.251
Hematological data					
White blood cell counts, 1000/μL	6.41 ± 1.88	7.38 ± 1.89	6.45 ± 1.57	5.39 ± 1.65	<0.001
Hemoglobin, g/dL	10.81 ± 1.35	11.07 ± 1.14	10.91 ± 1.44	10.43 ± 1.38	0.017
Platelets, 1000/μL	182.76 ± 64.28	232.48 ± 61.76	183.73 ± 46.95	131.29 ± 36.05	<0.001
Biochemical data					
Aspartate aminotransferase, U/L	21.22 ± 13.56	13.77 ± 7.01	20.00 ± 7.14	30.03 ± 18.00	<0.001
Alanine aminotransferase, U/L	20.42 ± 13.71	18.17 ± 12.30	19.77 ± 11.43	23.37 ± 16.57	0.084
Total bilirubin, mg/dL	0.30 ± 0.17	0.27 ± 0.14	0.31 ± 0.16	0.33 ± 0.21	0.106
Albumin, g/dL	4.03 ± 0.42	4.12 ± 0.36	4.08 ± 0.33	3.89 ± 0.52	0.003
Glucose, AC, mg/dL	112.48 ± 50.66	113.83 ± 45.91	111.65 ± 44.81	111.95 ± 60.64	0.965
Glycated hemoglobin A1c, %	7.56 ± 1.51	7.89 ± 1.23	7.66 ± 1.63	7.01 ± 1.60	0.183
Blood urea nitrogen before hemodialysis, mg/dL	65.29 ± 16.14	67.81 ± 13.97	67.02 ± 15.82	60.98 ± 17.79	0.029
Creatinine, mg/dL	10.72 ± 2.57	11.60 ± 2.88	10.88 ± 2.28	9.66 ± 2.16	<0.001
Uric acid, mg/dL	7.01 ± 1.39	7.22 ± 1.39	7.12 ± 1.24	6.67 ± 1.50	0.055
Sodium, mEq/L	137.77 ± 2.88	137.47 ± 2.71	138.21 ± 2.94	137.63 ± 2.99	0.300
Potassium, mEq/L	4.78 ± 0.69	4.95 ± 0.68	4.78 ± 0.66	4.60 ± 0.69	0.015
Corrected calcium, mg/dL	10.00 ± 1.01	10.16 ± 0.98	10.00 ± 1.06	9.82 ± 0.97	0.146
Phosphate, mg/dL	4.84 ± 1.44	5.36 ± 1.38	5.11 ± 1.35	4.04 ± 1.24	<0.001
Aluminum, g/dL	1.10 ± 1.17	1.17 ± 1.38	0.98 ± 0.71	1.16 ± 1.32	0.555
Iron, μg/dL	70.52 ± 30.22	69.61 ± 28.85	72.28 ± 33.28	69.47 ± 28.30	0.844
Total iron-binding capacity, μg/dL	262.03 ± 53.85	278.48 ± 51.75	260.88 ± 45.43	246.32 ± 60.43	0.005
Ferritin, ng/mL	246.68 ± 258.11	168.83 ± 174.81	230.80 ± 237.54	345.39 ± 318.45	0.001
Intact parathyroid hormone, pg/mL	297.47 ± 384.64	342.29 ± 286.70	351.61 ± 565.74	197.00 ± 170.87	0.035
High-sensitivity C-reactive protein, mg/L	5.86 ± 11.26	6.03 ± 9.62	5.50 ± 12.40	6.04 ± 11.74	0.952
Total cholesterol, mg/dL	162.61 ± 35.33	173.70 ± 36.46	165.67 ± 33.03	148.25 ± 31.91	<0.001
Triglyceride, mg/dL	133.34 ± 75.85	159.62 ± 89.82	123.35 ± 65.86	116.78 ± 62.66	0.002
High-density lipoprotein, mg/dL	45.23 ± 13.20	44.80 ± 14.10	47.77 ± 11.70	43.09 ± 13.45	0.121
Low-density lipoprotein, mg/dL	90.71 ± 29.46	96.98 ± 32.56	93.18 ± 27.90	81.82 ± 25.78	0.009

Patients who had diabetes mellitus were based on the American Diabetes Association criteria: A1c ≥ 6.5% or FPG ≥ 126 mg/dL (7 mmol/L), 2-h plasma glucose ≥ 200 mg/dL (11.1 mmol/L) during an OGTT, or in a patient with classic symptoms of hyperglycemia or hyperglycemic crisis with a random plasma glucose ≥ 200 mg/dL (11.1 mmol/L). In the absence of unequivocal hyperglycemia, diagnosis required two abnormal test results from the same sample or in two separate test samples. Patients who had hypertension were taking antihypertensive drugs regularly or their blood pressure was >140/90 mmHg at least twice. A1c: glycated hemoglobin; FPG: fasting plasma glucose; OGTT: oral glucose tolerance test.

**Table 2 diagnostics-14-02048-t002:** Cox regression analysis of 5-year MACEs in study patients according to basal variables and the FIB-4 score (*n* = 198; *p* > 0.05 excluded).

Significant Variables	Cox Univariate Analysis		Cox Multivariate Hazards Analysis	
	HR (95% CI)	*p*	aHR (95% CI)	*p*
Body weight after hemodialysis, kg (each increase of 1 kg)	1.038 (1.007–1.069)	0.015		
Diabetes mellitus, yes	7.297 (2.829–18.819)	<0.001	6.455 (2.466–16.893)	<0.001
Hemodialysis duration, year (each increase of 1 year)	0.925 (0.861–0.992)	0.030		
Urea reduction ratio (URR), % (each increase of 1%)	0.005 (0.000–0.342)	0.013		
KT/V Daugirdas	0.256 (0.082–0.801)	0.019		
Time-averaged concentration of blood urea nitrogen (TACurea), mmol/L (each increase of 1 mmol/L)	1.045 (1.005–1.086)	0.027	1.056 (1.010–1.104)	0.016
White blood cell counts, 1000/μL (each increase of 1000/μL)	1.301 (1.083–1.563)	0.005		
Total bilirubin, mg/dL (each increase of 1 mg/dL)	7.871 (1.345–46.071)	0.022	19.360 (2.619–143.086)	0.04
Glucose, AC, mg/dL (each increase of 1 mg/dL)	1.009 (1.003–1.016)	0.005		
Corrected calcium, mg/dL (each increase of 1 mg/dL)	0.550 (0.338–0.893)	0.016		
High-density lipoprotein, mg/dL (each increase of 1 mg/dL)	0.962 (0.925–1.000)	0.048		
FIB-4	1.222 (0.865–1.728)	0.256		

**Table 3 diagnostics-14-02048-t003:** Cox analysis of all-cause 5-year mortality in HD patients with chronic viral hepatitis (*n* = 198; *p* > 0.05 excluded).

Significant Variables	Cox Univariate Analysis		Cox Multivariate Hazards Analysis	
	HR (95% CI)	*p*	aHR (95% CI)	*p*
Age, year (each increase of 1 year)	1.064 (1.031–1.097)	<0.001		
Diabetes mellitus, yes	3.427 (1.703–6.893)	0.001	5.688 (2.358–13.720)	<0.001
Hemoglobin, g/dL (each increase of 1 g/dL)	0.542 (0.395–0.743)	<0.001	0.524 (0.369–0.745)	<0.001
Aspartate aminotransferase, U/L (each increase of 1 U/L)	1.021 (1.006–1.035)	0.005		
Total bilirubin, mg/dL (each increase of 1 mg/dL)	6.575 (1.425–30.344)	0.016		
Albumin, g/dL (each increase of 1 g/dL)	0.304 (0.192–0.481)	<0.001	0.538 (0.296–0.978)	0.042
Glucose, AC, mg/dL (each increase of 1 mg/dL)	1.007 (1.004–1.011)	<0.001		
Blood urea nitrogen before hemodialysis, mg/dL (each increase of 1 mg/dL)	0.968 (0.945–0.992)	0.009		
Creatinine, mg/dL (each increase of 1 mg/dL)	0.684 (0.585–0.800)	<0.001		
Sodium, mEq/L (each increase of 1 mEq/L)	0.862 (0.769–0.967)	0.011		
Potassium, mEq/L (each increase of 1 mEq/L)	0.430 (0.239–0.774)	0.005		
Corrected calcium, mg/dL (each increase of 1 mg/dL)	0.588 (0.398–0.868)	0.008		
Phosphate, mg/dL (each increase of 1 mg/dL)	0.689 (0.520–0.913)	0.010		
Total iron-binding capacity, μg/dL (each increase of 1 μg/dL)	0.992 (0.985–1.000)	0.049		
Ferritin, ng/mL (each increase of 1 ng/mL)	1.002 (1.001–1.003)	0.001		
High-sensitivity C-reactive protein, mg/L (each increase of 1 mg/L)	1.020 (1.002–1.038)	0.029		
Total cholesterol, mg/dL (each increase of 1 mg/dL)	0.988 (0.977–0.999)	0.026		
High-density lipoprotein, mg/dL (each increase of 1 mg/dL)	0.962 (0.933–0.993)	0.015		
FIB-4	1.683 (1.414–2.003)	<0.001	1.589 (1.262–2.001)	<0.001

## Data Availability

The raw data supporting the conclusions of this article will be made available by the authors on request.

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
