# Peer review of "Fibrosis-4 Score Is Associated with Mortality in Hemodialysis Patients with Chronic Viral Hepatitis: A Retrospective Study"

_diagnostics, 2024, doi:10.3390/diagnostics14182048_

Round 1

Reviewer 1 Report

Comments and Suggestions for Authors

Comments for authors

The manuscript is well written, and the authors showed new and relevant data related to major adverse cardiovascular events and mortality in patients with chronic viral hepatitis patients undergoing hemodialysis. These data highlighted the significant association between FIB-4 and 5-year all-cause mortality among these patients. However, there are minor issues that need to be addressed.

Minor revisions 

1) Page 2, line 72: In the Materials and Methods, it is necessary to clarify whether the 310 patients included in the study were all patients with chronic viral hepatitis infection undergoing hemodialysis under follow-up at this hospital during the study period.

2) Page 4, Figure 1: Each FIB-4 group included 66 patients. Was this random, or was there some criterion used?

3) Page 6, Table 2: Include the results of the multivariate Cox regression analysis (in the same way as in Table 3).

4) Page 5, line 160: What variables were used to adjust the multivariate Cox regression analysis for predictors for 5-Year MACE? This information should be included in the text or Table 2.

5) Page 6, line 185: What variables were used to adjust the multivariate Cox regression analysis for predictors of all-cause 5-year mortality? This information should be included in the text or Table 2.

Reviewer 2 Report

Comments and Suggestions for Authors

The study entitled "Fibrosis-4 Score Is Associated with Mortality in Hemodialysis Patients with Chronic Viral Hepatitis: A Retrospective Study" investigated whether the Fibrosis-4 score (FIB-4) is related to mortality in hemodialysis patients who also have hepatitis B and C. This retrospective study analyzes various patient data.

 Suggestions for making this article clearer include:

 1 Rationale: Patients undergoing hemodialysis with a high FIB-4 score likely have more severe conditions and a higher risk of mortality compared to those with a lower FIB-4 score.

2 Fibrosis Measurement: There are several methods for assessing fibrosis, as mentioned by the authors, ranging from invasive (biopsy liver) to non-invasive approaches, some of which are complicated. The authors discuss methods like MRE and TE, which are not convenient due to the need for expensive equipment. A simpler and more commonly used method by hepatologists is the APRI (AST to Platelets Ratio Index). However, in this article, APRI is referred to as the Age Platelet Ratio Index in line 47, which appears to be incorrect.

3 FIB-4 vs. APRI: Calculating FIB-4 is more complex than APRI, which is likely more popular. The study should demonstrate how FIB-4 is superior to APRI.

4 Retrospective of the Study: Although this is a retrospective study, the lab results mention that blood samples were collected, centrifuged, and stored at -70°C until testing (line 96). In practice, routine lab tests are typically performed immediately after blood collection. If the blood was stored at -70°C until use for the assays, this wouldn’t be considered a retrospective study. Therefore, the IRB (Institutional Review Board) needs to provide a clear explanation.

5 Patient Grouping and ROC Analysis: Patients were divided into three tertile groups to identify lab differences, but these groups weren’t used for ROC AUC analysis to determine the sensitivity and specificity of FIB-4. After obtaining sensitivity and specificity, a cut-off value should be established, and then the patients should be divided into two groups to analyze mortality. Generally, readers would prefer to know the likelihood of a patient with a specific FIB-4 score surviving for the next five years, as this would be more beneficial.

Round 2

Reviewer 2 Report

Comments and Suggestions for Authors

No additional comments.